# Colon cancer cell differentiation by sodium butyrate modulates metabolic plasticity of Caco-2 cells via alteration of phosphotransfer network

Ljudmila Klepinina[1]*, Aleksandr Klepinin[1], Laura Truu[1], Vladimir Chekulayev[1], Heiki Vija[2], Kaisa Kuus[3], Indrek Teino[3], Martin Pook[3], Toivo Maimets[3], Tuuli Kaambre[1]

1 Laboratory of Chemical Biology, National Institute of Chemical Physics and Biophysics, Tallinn, Estonia, 2 Laboratory of Environmental Toxicology, National Institute of Chemical Physics and Biophysics, Tallinn, Estonia, 3 Department of Cell Biology, Institute of Molecular and Cell Biology, University of Tartu, Tartu, Estonia

* ljudmila.ounpuu@gmail.com

**Data Availability Statement:** All relevant data are within the manuscript and its Supporting Information files.

## Abstract

The ability of butyrate to promote differentiation of cancer cells has important implication for colorectal cancer (CRC) prevention and therapy. In this study, we examined the effect of sodium butyrate (NaBT) on the energy metabolism of colon adenocarcinoma Caco-2 cells coupled with their differentiation. NaBT increased the activity of alkaline phosphatase indicating differentiation of Caco-2 cells. Changes in the expression of pluripotency-associated markers OCT4, NANOG and SOX2 were characterized during the induced differentiation at mRNA level along with the measures that allowed distinguishing the expression of different transcript variants. The functional activity of mitochondria was studied by high-resolution respirometry. Glycolytic pathway and phosphotransfer network were analyzed using enzymatical assays. The treatment of Caco-2 cells with NaBT increased production of ATP by oxidative phosphorylation, enhanced mitochondrial spare respiratory capacity and caused rearrangement of the cellular phosphotransfer networks. The flexibility of phosphotransfer networks depended on the availability of glutamine, but not glucose in the cell growth medium. These changes were accompanied by suppressed cell proliferation and altered gene expression of the main pluripotency-associated transcription factors. This study supports the view that modulating cell metabolism through NaBT can be an effective strategy for treating CRC. Our data indicate a close relationship between the phosphotransfer performance and metabolic plasticity of CRC, which is associated with the cell differentiation state.

## Introduction

Colorectal cancer originates from the epithelial cells lining the large intestine. A rapid cell turnover is important for the growth of normal colonic epithelium. Stem cells continuously

**Funding:** This work was supported by the institutional research funding IUT23-1 of the Estonian Ministry of Education and Research.

**Competing interests:** The authors have declared that no competing interests exist.

proliferate in the base of the colonic crypt producing transit cells that migrate toward the upper part of the crypt, acquire the differentiated phenotype and finally undergo apoptosis [1]. Increased alkaline phosphatase activity has been shown to indicate differentiation of colon and colon cancer cells [2–6]. Since cancer cells often lack features of terminally differentiated cells, it has been proposed that tumor cells might arise from undifferentiated stem/progenitor cells. Alternatively, cancer cells can undergo progressive dedifferentiation into cells with "stem-like" properties known as cancer stem cells (CSC) [7,8]. Moreover, CSCs might use the same molecular pathways to facilitate self-renewal as normal stem cells. The increased expression of self-renewal regulatory factors important in cell pluripotent state such as OCT4, NANOG and SOX2 was detected in various types of cancers [9,10]. In clinical studies, the expression of these stem cell markers has been found to correlate with poor differentiation, advanced disease stages and worse overall survival in various cancers [11]. The OCT4 gene can generate several transcripts and isoforms by alternative splicing and these are differentially expressed in pluripotent and non-pluripotent cells [12]. In general, OCT4A together with OCT4B1 are highly expressed in embryonic stem and embryonic carcinoma cells and down-regulated after differentiation, while OCT4B is mainly the transcript expressed in somatic tumor cells [12]. Besides, OCT4B1 has been also found to be highly expressed in gastric, bladder, brain and colorectal cancers [13–16]. The expression of OCT4B4, which sequence is very similar to OCT4B, has been reported in embryonic carcinoma cells [17]. Additionally, there are OCT4 pseudogenes, which have been shown to be expressed in cancers. Those transcripts together with the different splice variants from the main gene make it difficult to detect the OCT4 expression level [18,19]. Aberrant NANOG expression has been also reported in many types of cancer including germline tumors, breast, prostate and colorectal cancers [20–22]. There are two different NANOG transcripts that can be expressed from the main gene [23] together with the transcripts from NANOG gene tandem duplication (NANOGP1) and NANOG pseudogenes [24]. The expression of SOX2 has been associated with cancer stem-like properties in skin, bladder and colorectal cancers [25–27].

The differentiation of colonic epithelial cells relies on numerous determinants including growth factors, hormones, vitamins as well as intestinal microbiota-derived metabolites such as short-chain fatty acids (SCFAs) [28]. Butyrate, which is present at relatively high concentrations (mM) in the colon lumen, is one of the most abundant SCFA produced by bacterial fermentation of dietary fibers. This metabolite exerts multiple beneficial effects on the intestinal homeostasis. Being the main energy source for normal colonocytes, butyrate also promotes proliferation of colonic epithelial cells, modulates immune and inflammatory responses, improves intestinal barrier function, and stimulates mucus secretion increasing vascular flow and motility [29]. More recently, attention has been given to butyrate due to its ability to suppress the *in vitro* proliferation of colon carcinoma cells. The anti-carcinogenic effect of butyrate has been attributed to its function as a histone deacetylase (HDAC) inhibitorIn various cancer cell lines, HDAC inhibitors suppress cell proliferation via cell cycle arrest, induce differentiation and apoptosis, reduce angiogenesis and modulate immune response [30].

The opposing effects of butyrate on the proliferation of normal versus cancerous colon cells has been explained by the "Warburg phenotype" of cancer cells [31]. Normally differentiated colon cells rely primarily on mitochondrial oxidative phosphorylation (OXPHOS) for energy production. Butyrate is metabolized in the β-oxidation pathway to acetyl-CoA, which enters the tricarboxylic acid (TCA) cycle and is used for ATP generation. The tendency of cancer cells to enhance glucose metabolism through aerobic glycolysis (Warburg effect) leads to diminished use of OXPHOS and butyrate as a substrate. Unmetabolized butyrate accumulates in the cell nucleus and functions as an HDAC inhibitor to control genes that inhibit cell proliferation and increase apoptosis [31].

Several studies demonstrated that butyrate could affect cancer cell proliferation by acting on individual enzymes of the glycolytic and OXPHOS pathways. Butyrate has been shown to increase oxidative pathway and/or decrease glycolytic metabolism in lung tumor cells (H460), colorectal adenocarcinoma cells (HT29, Caco-2, HCT116) and breast cancer cells (MCF-7, T47-D, MDA-MB231). This was correlated either with suppressed proliferation or induced differentiation or both together [32–36].

Although much is known about the rearrangements in energy producing pathways that occur during malignant transformation, the way how the energy is being transported within the cancer cell remains largely undiscovered. Creatine kinase (CK) and adenylate kinase (AK) that transfer phosphoryl groups between creatine phosphate, ADP, ATP and AMP are considered to facilitate the intracellular energetic communication. CK and AK phosphotransfer networks are defined as circuits of enzymes catalyzing sequential series of reversible transphosphorylation reactions linking ATP production and consumption sites in the cell (reviewed in [37]). It is noteworthy to mention that the phosphotransfer network of cell undergoes profound alterations during cancer development. Our previous studies revealed elevated AK activity and downregulated CK network in several tumors including human colorectal and breast cancer tissues, mouse neuroblastoma (Neuro-2A) and human embryonal carcinoma cells [38–40]. It has been proposed that diminished CK network can be partly compensated by other phosphotransfer enzymes, such as AK and glycolytic networks [41]. An important role for AK is emerging in cancer. AK4 has been reported to promote lung cancer progression and metastasis as well as modulate anti-cancer drug sensitivity in HeLa cells [42,43]. Recently, AK6 was proposed to be a potent modulator of metabolic reprogramming by regulating lactate dehydrogenase A (LDHA) activity in colon cancer stem cells [44]. Moreover, different AK isoforms have a prognostic biomarker potential for various cancer types (S1 Table).

Since SCFAs can alter the cellular metabolism of cancer cells, we hypothesized that treatment with sodium butyrate (NaBT) may reverse cancer-induced changes in phosphotransfer network of colon adenocarcinoma (Caco-2) cells including AK pathway. In addition to evidence supporting this hypothesis, we found that the flexibility of phosphotransfer networks depends on the availability of key metabolic substrates. The differentiation of Caco-2 cells was determined by increased alkaline phosphatase activity. In addition, NaBT-treatment resulted in the enhanced oxidative metabolism along with the changes in gene expression of the main pluripotency-associated transcription factors. Altogether, this indicates the link between regulation of phosphotransfer system and metabolic plasticity of cancer cells associated with the cell differentiation state.

## Materials and methods

### Patients and tissue samples

Human colorectal cancer and adjacent normal tissues were obtained from eight colorectal cancer patients between 55–87 years old, who underwent surgery at the North Estonia Medical Centre (Tallinn, Estonia). The adjacent normal tissue specimens were collected from an incision > 5 cm away from the carcinoma sites. Immediately after surgery, tissue samples were collected into RNAlater solution, frozen in the liquid nitrogen and kept at -80˚C.

The pathological information of all patients was obtained from the Oncology and Hematology Clinic of the North Estonia Medical Centre. All patients examined had primary tumors and had not received prior radiation or chemotherapy. The study was approved by the Medical Research Ethics Committee (National Institute for Health Development, Tallinn) and conducted in compliance with the Declaration of Helsinki and the European Convention on Human Rights and Biomedicine. Written informed consent was obtained from all patients.

## Cell culture and cell lines

The established Caco-2 cell line was obtained from American Type Tissue Culture Collection; HTB-37[TM]; p5). Cells were maintained in Eagle's minimum essential medium (Corning), supplemented with 10% fetal bovine serum (Gibco), 1% non-essential amino acids 100X (Corning), 100 U·ml[-1] penicillin (Gibco), 100 μg·ml[-1] streptomycin (Gibco) and 50 μg·ml[-1] gentamicin (Gibco) at 37˚C in a humidified incubator supplied with 5% $CO_2$. Cells were sub-cultured every 3 days by mild trypsinization and seeded at a density of 20 000 cells·cm[-2]. The viability of cells was around 95% according to trypan blue exclusion test. Cells were counted using a Bürker-Türk counting chamber. The maximal passage number of the cells used in the study was 20 and PCR based mycoplasma testing (see Supplementary) was used to confirm no mycoplasma contamination in the cell culture.

Human embryonic stem cells (WA09, National Stem Cell Bank, WiCell; received Feb-2020, p25) were cultured on 6-well tissue culture plates (BD Biosciences) coated with Matrigel (BD Biosciences) in mTeSR1 media (STEMCELL Technologies) according to the manufacturer's specifications. The culture medium was changed daily. Cells were passaged mechanically with micropipette tip after 3–4 days and cultured in the presence of 5% $CO_2$ at 37˚C in humid conditions. The maximal passage number of the cells used in the study was 70 and the normal karyotype of the cells was routinely confirmed by G-banding.

## Cell viability and cytotoxicity assay

Cells were grown on 6 and 96-well plates. 24 h after plating, cells were incubated with various concentrations of NaBT (Acros Organics) for 24, 48 or 72 h. NaBT solution was prepared immediately before use by dissolving NaBT powder in complete growth medium. After each treatment, cell viability was assessed using MTT assay as described previously [45] and trypan blue test [46]. The cytotoxicity was assayed by lactate dehydrogenase (LDH) release after NaBT treatment. LDH activity was measured in both floating dead cells and viable adherent cells by using a standard kinetic determination [47]. The cytotoxicity was calculated as LDH activity in culture medium divided by total LDH activity.

## Quantitative RT-PCR, semi-quantitative RT-PCR and enzymatic digestion

RNA from Caco-2 cells, frozen human colorectal cancer and adjacent normal samples was extracted with the RNeasy Mini Kit (QIAGEN Sciences) followed by DNase I treatment (Thermo Fisher Scientific). Synthesis of cDNA was performed with RevertAid First Strand cDNA Synthesis Kit using oligo(dT)$_{18}$ primer (Thermo Fisher Scientific). Quantitative RT-PCR was performed using Maxima SYBR Green/ROX qPCR Master Mix (Thermo Fisher Scientific), specific primers (S2 and S4 Tables) and real-time PCR machine LightCycler 480 (Roche). Semi-quantitative RT-PCR was carried out with FIREPol Master Mix (Solis BioDyne) and gene-specific primers (S4 and S5 Tables). PCR products were analyzed on 2% agarose gel electrophoresis. Images were obtained and band intensities were quantified with Biospectrum 510 Imaging System (UVP, LLC). To distinguish between OCT4A and its pseudogene expression, the approximately 500 bp band was excised from agarose gel, followed by purification of DNA with FavorPrep GEL/PCR Purification Mini Kit (Favorgen). Recovered DNA was subjected to 5 units of ApaI (Thermo Fisher Scientific) digestion overnight followed by separation on 2% agarose gel electrophoresis and imaging. Transcripts were identified by size, based on the previously published results with these primers. The respective areas were selected as rows to summarize the results for different transcripts in the figures.

## High-resolution respirometry

Mitochondrial respiration was measured in intact cells by means of high-resolution respirometry using the Oroboros® Oxygraph-2K (Oroboros Instruments, Innsbruck, Austria). The DATLAB 5 software (Oroboros Instruments) was used for real-time data acquisition and subsequent data analysis. Measurements were performed at 25˚C in the respiration medium comprising 110 mM sucrose, 60 mM K-lactobionate, 0.5 mM EGTA, 3 mM $MgCl_2 \cdot 6H_2O$, 20 mM taurine, 3 mM $KH_2PO_4$, 20 mM HEPES, adjusted to pH 7.1 with KOH and supplemented with 2 $mg \cdot ml^{-1}$ bovine serum albumin free from essential fatty acids (Sigma), 0.5 mM DTT and 9.6 µg/ml leupeptin. The rates of oxygen consumption were normalized to cell number (nmol of $O_2 \cdot min^{-1} \cdot 10^{-6}$ cells). To measure the oxygen consumption through different segments of the respiratory chain and analyze basic respiratory properties, cells were treated with the combination of substrates, uncouplers and inhibitors of respiration as depicted in S1 Fig.

## Mitochondrial membrane potential

Caco-2 cells were seeded into black clear-bottom 96-well plates (Greiner Bio-One) at a density of 3 000 cells per well. The next day the growth medium was changed to the medium containing 1 mM NaBT and cells were incubated for 48 h at 37˚C in a humidified incubator supplied with 5% $CO_2$. After treatment, the mitochondrial membrane potential was measured using the fluorescent dye tetramethylrhodamine methyl ester (TMRE). Cells were incubated with 20 nM TMRE in complete growth medium at 37˚C for 30 min protected from light. The medium was removed, and cells were washed 3 times with PBS to remove background fluorescence from the cell culture medium. 100 µl of PBS was added into each well, and fluorescence intensities were measured on a FLUOstar Omega microplate reader (BMG Labtech) (excitation: 550 nm and emission: 590 nm). In order to exclude any possible impact of the plasma membrane potential on the fluorescence intensities, each assay was performed in parallel as above plus 10 µM of FCCP, which disrupts the mitochondrial membrane potential. After measurements, PBS was removed, and cells were lysed with 1% SDS solution in PBS. Protein concentrations were determined by Pierce BCA Protein Assay Kit according to the manufacturer recommendations. All data were expressed as the total TMRE fluorescence minus FCCP-treated TMRE fluorescence and normalized to protein content.

## Immunofluorescence analysis

Cells were seeded onto glass coverslips at a density of $1.7 \cdot 10^4$ $cells \cdot cm^{-2}$ in 12-well plates (Greiner Bio-one) overnight and treated with 1 mM NaBT for 48 h. For mitochondria staining, 200 nM MitoTracker Red CMXRos (Invitrogen) was added into culture medium for 30 min before fixing with 4% paraformaldehyde in PBS for 10 min at room temperature (RT). Fixed cells were washed with PBS and permeabilized with 0.1% Triton X-100 in PBS for 10 min. After washing with PBS, the coverslips were incubated with anti-beta tubulin antibody (ab6046, Abcam) in 2% BSA/PBS overnight at 4˚C. Thereafter, these were washed with PBS, followed by incubation with Alexa Flour 488 conjugated anti-rabbit antibody (ab96899, Abcam) for 1 h at RT. After washing 3X with PBS, the coverslips were mounted with ProLong Gold antifade reagent containing 4',6-dia-midino-2-phenylindole dihydrochloride (DAPI,Thermo Fisher Scientific). Cells were examined under Olympus FluoView FV10i-W inverted laser scanning confocal microscope.

## Preparation of cell lysates

The cell pellets were re-suspended in ice-cold lysis buffer containing 20 mM MOPS pH 8.0, 20 mM $MgCl_2 \cdot 6H_2O$, 10 mM glucose, 200 mM NaCl, 10 mM EDTA-$Na_2$, 0.25% Triton-X, and protease inhibitor cocktail (Roche). Cell lysates were homogenized using a Retsch Mixer Mill (Retsch) at 25

Hz for 2 min and centrifuged at 12,000 rcf for 20 min at 4˚C. The supernatants were used for enzymatic assays. Protein concentrations were determined by Pierce BCA Protein Assay Kit according to the manufacturer recommendations using bovine serum albumin (BSA) as a standard.

## Enzymatic activity assays

Enzymatic activities were measured spectrophotometrically in cell lysates at 25˚C with a FLUOstar Omega microplate reader. Alkaline phosphatase (ALP) activity was determined with *p*-nitrophenyl phosphate (pNPP) as a substrate using a modification of the published method [48]. The assay mixture contained 0.1 M Tris-HCl pH 9.8, 5 mM $MgCl_2$ and 2 mM pNPP. The activity was associated with a decrease in absorbance at 405 nm. Specific activity was calculated by using an extinction coefficient 6.22 $mM^{-1}cm^{-1}$. Citrate synthase (CS) activity was determined by measuring the rate of thionitrobenzoic acid production at 412 nm [49]. Hexokinase (HK) activity was determined using a standard glucose-6-phosphate dehydrogenase (G6PDH)-coupled spectrophotometric assay [50]. Lactate dehydrogenase and pyruvate kinase activities were quantified by measuring the decrease in absorbance of NADH at 340 nm [47,51]. CK and AK activities were measured by an enzyme-coupled spectrophotometric assay in the direction of ATP formation as described earlier [52,53]. All enzymatic activities were normalized per mg of cell protein.

## Nucleotide measurements

ATP level and ATP/ADP ratio were determined by ultra-performance liquid chromatography (UPLC) as described previously [54] with some modification. Briefly, after quick growth medium removal, cells were rinsed with saline solution, quenched with 0.6 M ice-cold $HClO_4$ and immediately frozen in liquid nitrogen. After cell collection, samples were centrifuged at 10,000 rcf for 5 min at 4˚C. The supernatant was neutralized with 2 M $KHCO_3$ and the pellet was used in protein assay. To remove salt, samples were centrifuged at 10,000 rcf for 15 min at 4˚C. Nucleotides in the supernatant were separated on a reversed-phase C18 column Separon SGX 5 µm 3x150 mm (Tessek, Czech Republic) with the Waters Acquity UPLC equipped with a PDA detector. A phosphate buffer (100 mM) with tetrabutylammonium sulfate (10 mM) and methanol mixture (water:methanol; 60:40% (v:v)) was used as mobile phase at 0.6 $mL \cdot min^{-1}$ flow rate. Gradient elution was applied and during 9 min all nucleotides were completely separated. To estimate ATP production through mitochondrial respiration and glycolysis, we measured ATP/ADP ratio in the presence of OXPHOS inhibitors (rotenone, antimycin A and oligomycin 1 µg/mL) or glycolysis [2-deoxyglucose (DOG) 6mM], as described previously [55]. The ATP levels were normalized per mg of cell protein.

## Statistical analysis

Data analysis was performed using SigmaPlot software. Results are expressed as means ± SEM. Statistical comparisons between control group and treated cells were made by two-tailed Student's t test for unpaired samples. To assess differences between multiple groups, a 2-way analysis of variance (ANOVA) was applied. The Turkey test was used for post-hoc analysis. The statistical analysis was conducted at 95% confidence level. A P value less than 0.05 was considered statistically significant.

# Results

## Incubation of cells with 1 mM NaBT for 48 h induced cell differentiation without any toxic effect

In this study, we evaluated a possible relationship between cellular differentiation and energy metabolism in colon cancer cells. To avoid any interferences caused by cytotoxicity, we first

validated the best experimental conditions to study effects of NaBT on cellular bioenergetics. Caco-2 cells were treated with various concentrations of NaBT for 24, 48 and 72 hours. The viability of metabolically active cells was estimated by MTT assay (Fig 1A). The viability of the treated cells is presented relative to that of the untreated cells (control), which is regarded as 100% cell viability. In addition, the cytotoxic effect of NaBT was assayed by lactate dehydrogenase (LDH) release. As expected, treatment with NaBT induced time and dose-dependent inhibition of cell growth (Fig 1A). Treatment of cells for 24 hours was not enough to reveal inhibitory effect of NaBT on the cell growth since the doubling time of Caco-2 monolayer cell culture is approximately 32 hours (ATCC). Gradual inhibition of cell growth was observed after 48 hours of incubation. The viability of cells was 90±2% with 1 mM NaBT, 73±7% with 2 mM, 56±10% with 5 mM and 37±3% with 10 mM. Longer incubation with NaBT (72 hours) resulted in more prominent effect on the viability. Specifically, cell viability of 85±6%, 68±9%, 39±5% and 17±4% were observed for 1, 2, 5 and 10 mM NaBT, respectively. The cytotoxic effect of NaBT appeared after 48 h of treatment at concentrations above 2 mM (Fig 1B and S2 Fig). When the incubation was extended to 72 hours, the cytotoxicity of NaBT was also observed at 2 mM concentration. In contrast, 1 mM NaBT was non-toxic for Caco-2 cells at all studied time points.

The inhibition of cell growth was also accompanied by increased ALP activity (Fig 1C), a well-established marker of colon cell differentiation [56]. After 48 h of incubation with 1 mM NaBT, the ALP activity increased almost 10 times compared with untreated control. Furthermore, treated cells resembled epithelial-like cells with a polygonal shape and had more regular dimensions compared to cells cultured in the absence of NaBT (Fig 2). Thus, the incubation of cells with 1 mM NaBT for 48 h was sufficient to induce cell differentiation without any notable toxic effect, and this treatment protocol was used in subsequent experiments.

## Treatment with NaBT affected expression of stem cell markers OCT4 and SOX2 but not NANOG

As treatment with 1mM NaBT for 48 h induced changes in Caco-2 cell growth and increased ALP activity, pointing to cancer cell differentiation, we decided to analyze the expression pattern of main pluripotency-associated stem cell markers to indicate the presence of putative cancer stem cells and also characterize the possible splice variants of OCT4 and NANOG which expression could be modulated by the epigenetic changes induced by NaBT. The expression of OCT4, NANOG and SOX2 was detected in Caco-2 cells at mRNA level (S3 Fig) and the change in relative expression was semi-quantitatively measured from band intensity levels (Fig 3). To examine the expression pattern of different splice variants of OCT4 in Caco-2 cells, we used primers that were previously described to specifically amplify OCT4A, OCT4B and OCT4B1 [12]. In order to verify the expression of OCT4A, we used restriction analysis with ApaI for the amplified OCT4A product as only the true OCT4A product contains this restriction site [57]. Interestingly, OCT4A transcript was found to be expressed only after treatment with NaBT and the control cells did mainly express transcripts from OCT4 pseudogenes that were non-specifically amplified with the previously published OCT4A primers (S3B Fig). In order to confirm the expression changes for OCT4A we designed new primers, which specifically detected OCT4A. With the subsequent quantitative RT-PCR analysis we confirmed increased OCT4A expression after NaBT treatment (Fig 4). Additionally, we detected the expression of OCT4B1 variant, while OCT4B variant was missing (S3A and S5 Figs). In human embryonic stem cells (hESCs, cell line WA09), OCT4B appeared as a double band (S3A Fig) where the shorter product points to the existence of OCT4B4 transcript [17] that was also detected from Caco-2 cells (S5 Fig). Semi-quantitative densitometric analysis of

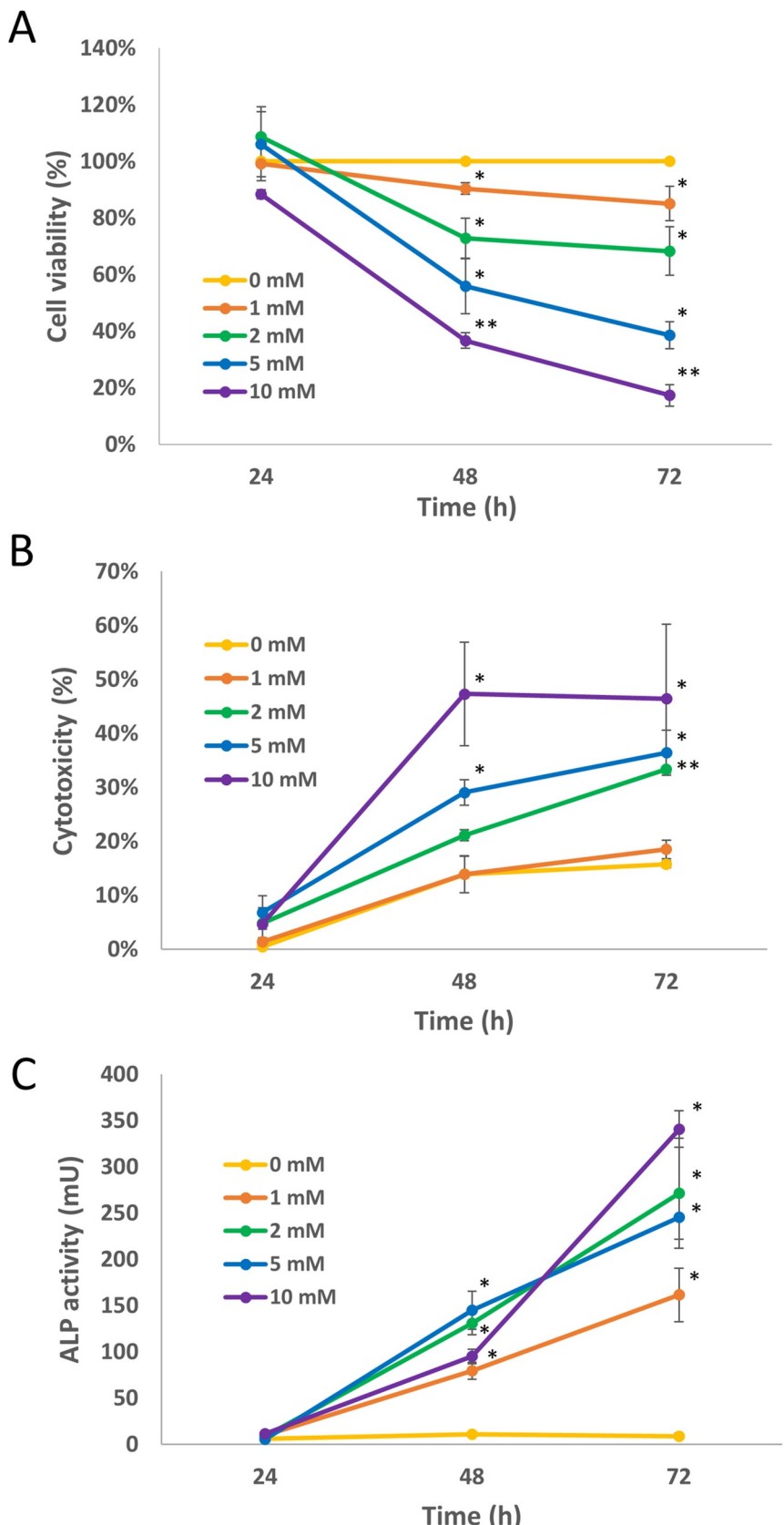

**Fig 1. Caco-2 cell culture sensitivity and differentiation response to sodium butyrate (NaBT).** Cells were incubated in the presence of various concentrations of NaBT for 24, 48 or 72 hours. (A): The viability of cells was analyzed by MTT assay. (B): Cytotoxicity was estimated by measurement of lactate dehydrogenase release after treatment with NaBT. (C) Alkaline phosphatase (ALP) activity assay was used to estimate differentiation status of the cells. Data are presented as mean ± SEM (n = 3–5). *P < 0.01; **p < 0.001 (ANOVA followed by Tukey post-hoc test).

OCT4B4 band intensities revealed that this transcript is more expressed after treatment with NaBT (Fig 3). Unfortunately, we were not able to confirm this by quantitative RT- PCR due to lack of suitable specific primers. We also checked for NANOG expression in hESCs and Caco-2 cells (S3A Fig). Although both transcripts from NANOG (NANOG1, NANOG2) were detected in Caco-2 cells, the expression of NANOG2 was prevailing and no differences were noted after treatment with NaBT. In embryonic stem cells, only NANOG1 transcript was observed. Interestingly, we did not detect changes in the expression of SOX2 after differentiation of Caco-2 cells with NaBT (Figs 3 and 4).

Next, we were interested to find out which variants of the analyzed stem cell markers are expressed in primary colorectal tumors. We used colorectal tumor tissue samples together with the control samples taken from the surrounding normal tissue of the same patient and characterized the expression of the same pluripotency-associated stem cell markers in this material (S4A Fig). The general OCT4A primers amplified a product that was more abundant in some of the analyzed tumor samples compared with normal tissue, but at the same time there were samples that did not show this tendency. Verification of OCT4A with ApaI restriction analysis showed that most of the amplified products (excluding sample 6) were OCT4 pseudogenes and together with semi-quantitative measurements of this band and quantitative RT-PCR with OCT4A specific primers we can suggest the increased expression of OCT4 pseudogenes in colorectal cancer (Figs 3, 4 and S4B). OCT4B1 expression was detected showing no

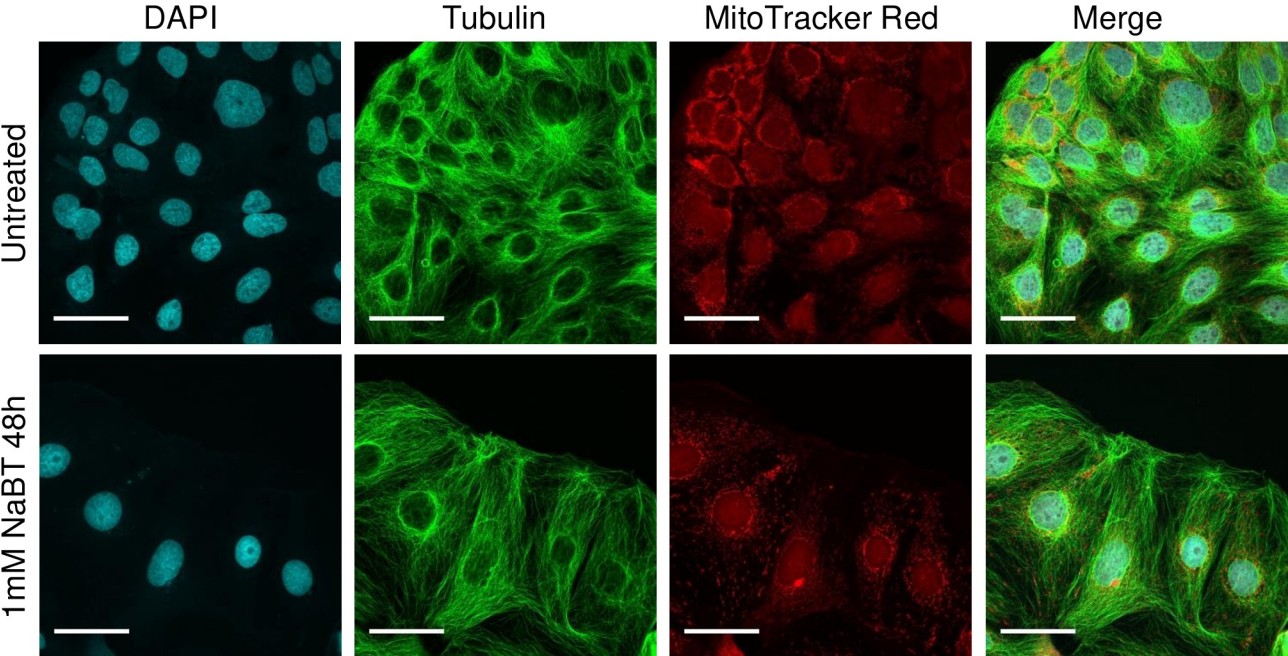

**Fig 2. Confocal microscopy of untreated and sodium butyrate (NaBT)-treated Caco-2 cells.** Morphological changes of cells occur after treatment of cells for 48h with 1mM NaBT. Cells were stained with MitoTracker (red), anti-whole tubulin (green) and DAPI (blue). For all the above, representative images are shown. Scale bars: 50 μm.

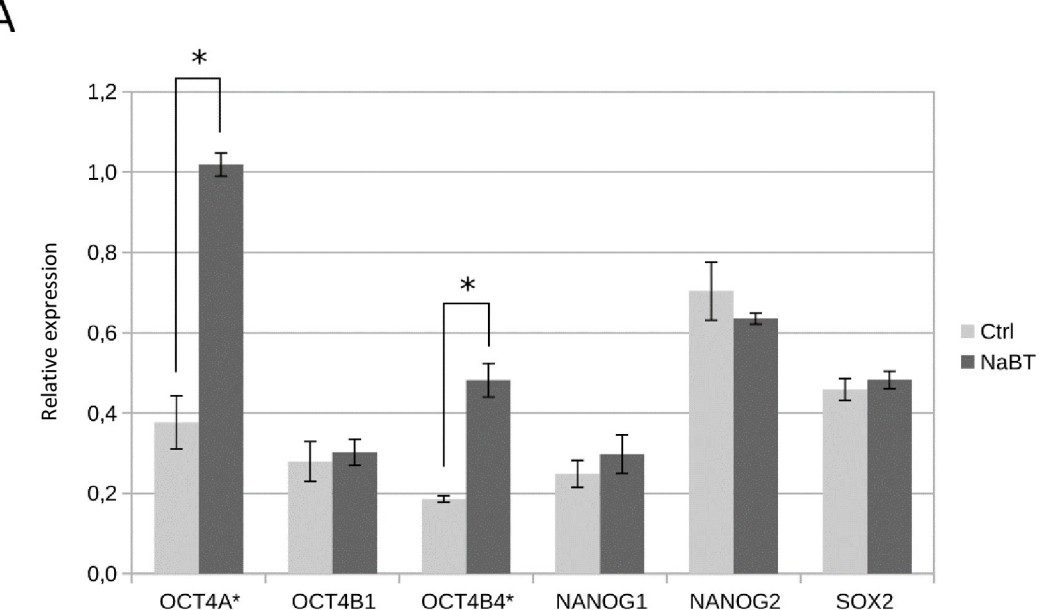

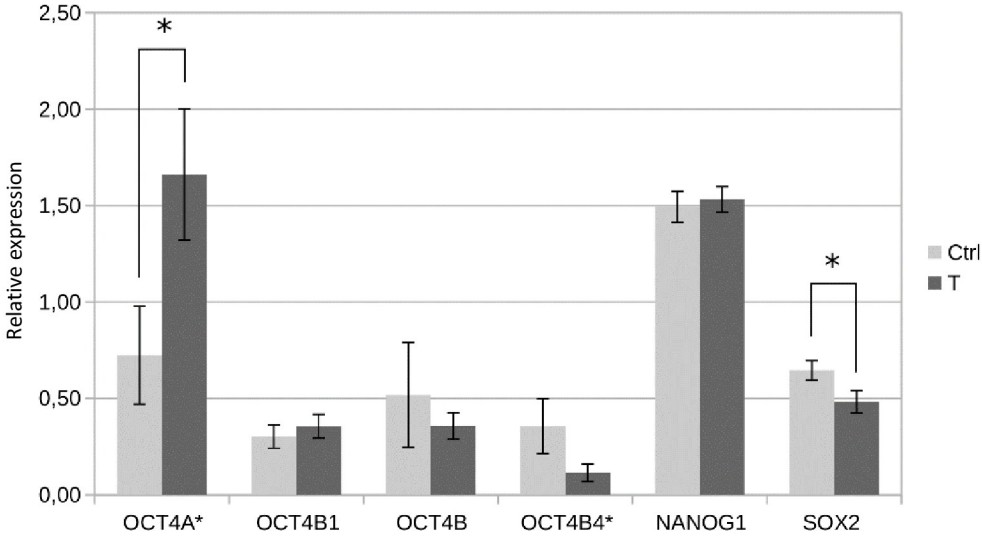

**Fig 3. Semi-quantitative densitometric analysis of pluripotency-associated transcription factors.** (A): Densitometric band intensity measurements from S3A Fig showing the relative changes between NaBT-treated (1mM, 48h) Caco-2 cells (NaBT) and without treatment (Ctrl) (B): Densitometric band intensity measurements from S4A Fig showing relative changes between primary colorectal tumor samples (T) and adjacent normal tissue samples (Ctrl). Data are presented as mean ± SEM ($^*$p < 0.05, Student's t test).

evident changes between tumor and normal tissue samples (Fig 3) and OCT4B together with OCT4B4 was showing variable expression pattern between the analyzed patient tissue samples (Fig 3 and S4A Fig). Interestingly, we did not see expression of NANOG transcript variant 2 (NANOG2) in primary tissue samples, while the expression of variant NANOG1 was present

A

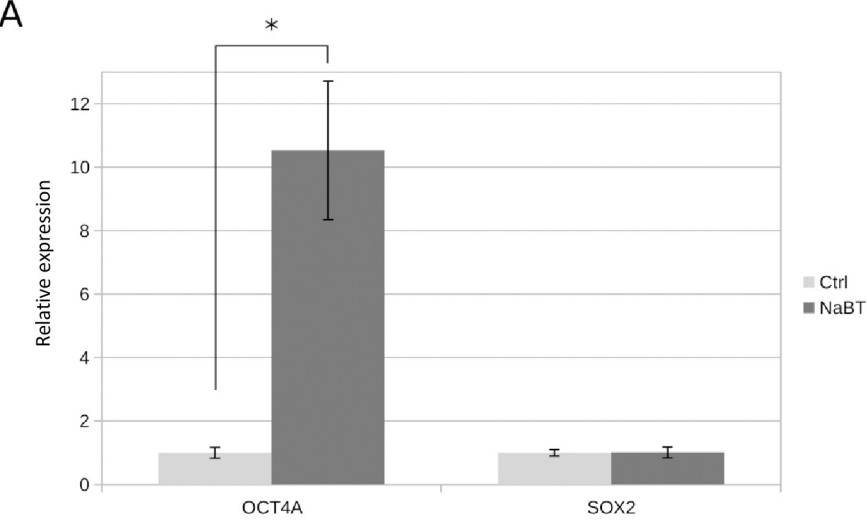

B

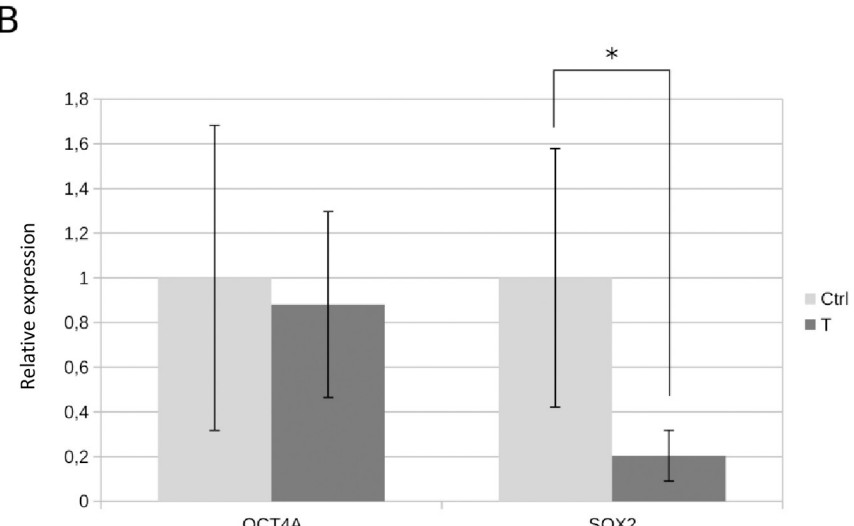

**Fig 4. Quantitative RT-PCR analysis of pluripotency-associated transcription factors OCT4A and SOX2.** (A): Relative gene expression in Caco-2 cells treated for 48h with 1 mM sodium butyrate (NaBT) or without treatment (Ctrl). (B): Relative gene expression in primary colorectal tumor samples (T) and adjacent normal tissue samples (Ctrl). Data are presented as mean ± SEM ($*p < 0.05$, Student's t test).

without evident changes in all the samples (Fig 3 and S4A Fig). Surprisingly, the expression of SOX2 was more detected in normal samples and most tumor tissue samples had less SOX2 expressed than in adjacent normal tissue (Figs 3, 4 and S4A).

## Treatment of cells with NaBT increased oxidative metabolism

Caco-2 cells displayed mixed profile of small punctate and elongated mitochondria (Fig 2). Treatment of cells with 1 mM NaBT did not induce any drastic changes in mitochondria morphology, total mitochondrial mass or mitochondrial membrane potential (Fig 2 and S6 Fig).

To investigate whether NaBT affects mitochondria on the functional level, we measured the rates of oxygen consumption by Caco-2 cells using high-resolution respirometry (S1A Fig).

The ability of cells growing with or without NaBT to consume oxygen was measured in the absence or presence of NaBT in the respiratory medium (Fig 5A and 5B). We also used substrates specific for the mitochondrial complex I (5 mM glutamate and 2 mM malate) and complex II (10 mM succinate) to assess the impact of individual electron transfer system components on the overall respiratory performance of cells. In the absence of butyrate in the respiratory medium, treated cells and cells growing without NaBT displayed similar rates of oxygen consumption (Fig 5A). However, the supplementation of respiratory medium with 5 mM NaBT resulted in significantly increased respiration rates only in 48h NaBT pre-treated cells with each substrate tested suggesting that butyrate is used as a substrate for an oxidative metabolism in more differentiated colon cancer cells (Fig 5B).

To analyze the impact of NaBT on the key parameters of mitochondrial respiration, we measured oxygen consumption rates in non-permeabilized Caco-2 cells using mitochondrial stress test protocol (S1 Fig). No significant difference was observed in the routine respiration, proton leak or ATP-coupled respiration (Fig 5C). However, the maximal respiration rate following addition of uncoupling agent (FCCP) and spare respiratory capacity were significantly higher in NaBT-treated cells compared with untreated control cells, indicating that treated cells are more adaptable to cellular stress. Furthermore, the increased non-mitochondrial residual oxygen consumption (ROX) after treatment of cells with NaBT may indicate that NaBT can activate alternative oxidases or promote other oxygen consuming processes in cell cytosol.

To estimate the contribution of glycolysis and oxidative phosphorylation to ATP production, we measured the amount of ATP and the ratio of ATP/ADP in cells treated with inhibitors of OXPHOS (rotenone, antimycin A and oligomycin) or glycolysis [2-deoxyglucose (DOG)] (Fig 5D and 5E). The total ATP level as well as the ratio of ATP/ADP did not differ between NaBT-treated and untreated control cells. We found that in both experimental groups, most of ATP was generated by glycolytic pathway. However, the level of ATP produced by OXPHOS was increased and the amount of ATP generated by glycolysis was slightly decreased after incubation of cells with NaBT (Fig 5E). Namely, untreated cells produced 33 ±1% of total ATP by OXPHOS, while 67±1% of ATP was derived from glycolysis. After treatment with NaBT, 44±3% and 56±2% of ATP was produced by OXPHOS and glycolysis, respectively. In addition, enzyme assay study showed that the activities of glycolytic enzymes lactate dehydrogenase and pyruvate kinase were decreased in NaBT-treated cells (Fig 6A–6D). Altogether, these results suggest that differentiation of Caco-2 cells with NaBT induced a shift from glycolytic to more oxidative metabolism.

## Rearrangement of phosphotransfer system in response to NaBT treatment and changing microenvironmental conditions

To test whether there is a relationship between NaBT-induced changes in energy metabolism and a phosphotransfer system of cells, we measured the specific activities of CK, AK and glycolytic enzymes (Figs 6 and 7). As glutamine can be an alternative energy source for tumor cells (Fig 7A), we evaluated the changes in phosphotransfer system also in the glutamine-free media.

The reduced levels of glycolysis-derived ATP suggested that treatment of cells with NaBT might lead to disturbances in the glycolytic pathway. To investigate this, the activity of HK, an important regulator of earlier stages of glycolytic flux, was measured. The middle stage of glycolysis was evaluated by pyruvate kinase (PYK) activity and the end phase was estimated by LDH activity. The HK activity remained the same in all experimental groups (Fig 6E and 6F). However, both PYK and LDH activities were significantly decreased after treatment of cells

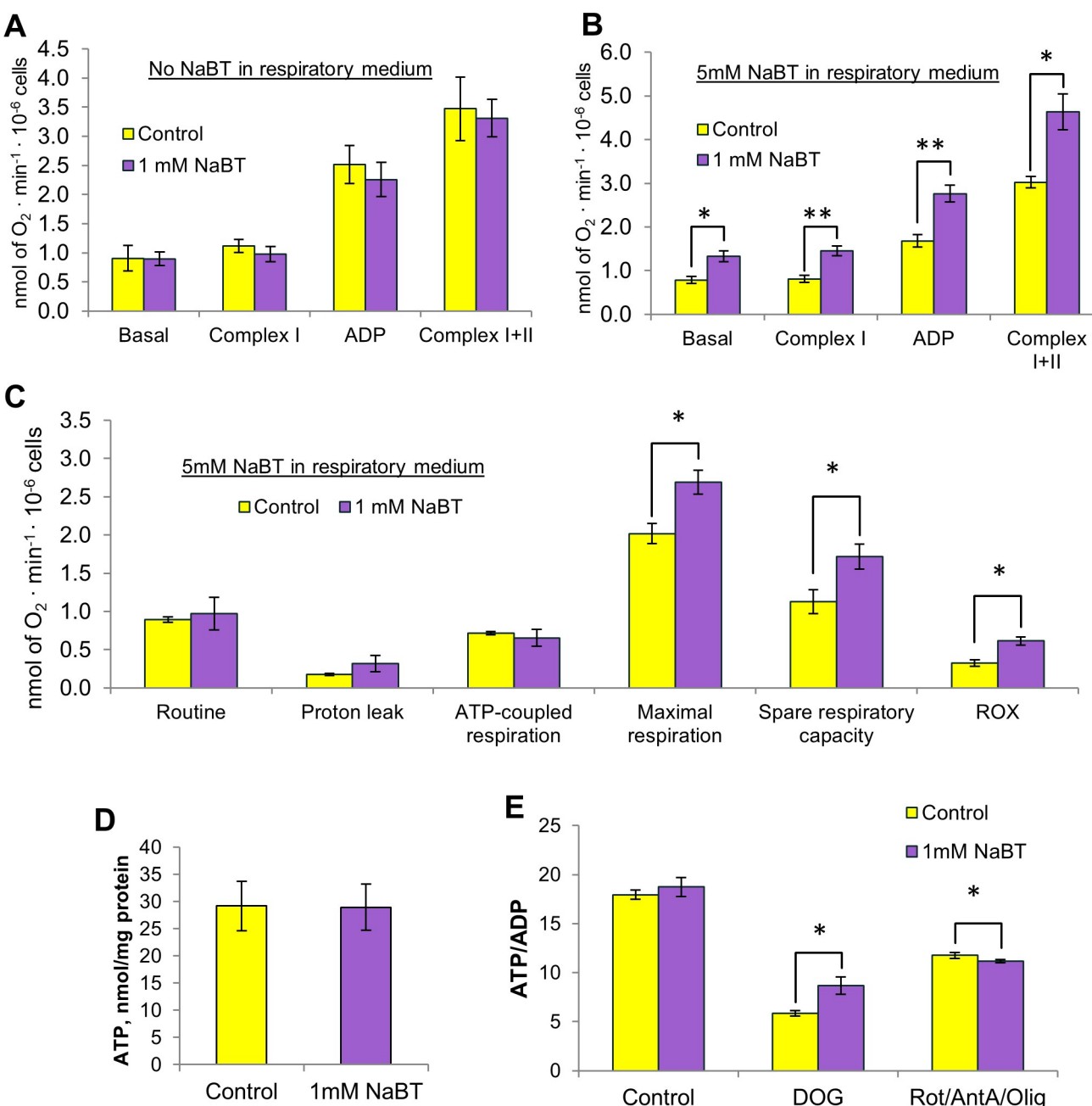

**Fig 5. 48-hour pre-treatment with sodium butyrate induced an increase in oxidative metabolism of Caco-2 cells.** (A): Oxygen consumption rates in the absence of sodium butyrate in the respiratory medium were measured using high-resolution respirometry. (B): Oxygen consumption rates in the presence of sodium butyrate. (C): Parameters of mitochondrial respiration obtained using mitochondrial stress test protocol. (D): ATP concentration measured by UPLC. (E): Effect of glycolysis and OXPHOS inhibition on the ATP/ADP ratio. All data are presented as mean ± SEM (n = 3–5; *p < 0.05, Student's *t* test). AntA–antimycin A, CS–citrate synthase, DOG– 2-deoxyglucose, NaBT–sodium butyrate, Olig–oligomycin, Rot–rotenone.

with NaBT in the presence of glutamine in the growth media (Fig 6A–6C). Interestingly, PYK activity did not change after NaBT-treatment in glutamine free media. In contrast, LDH activity was decreased after NaBT treatment in glutamine-free media containing lower levels of glucose (5 mM), while supplementation of glutamine-free media with higher glucose concentration (25 mM) resulted in similar levels of LDH activity in treated and untreated cells. These

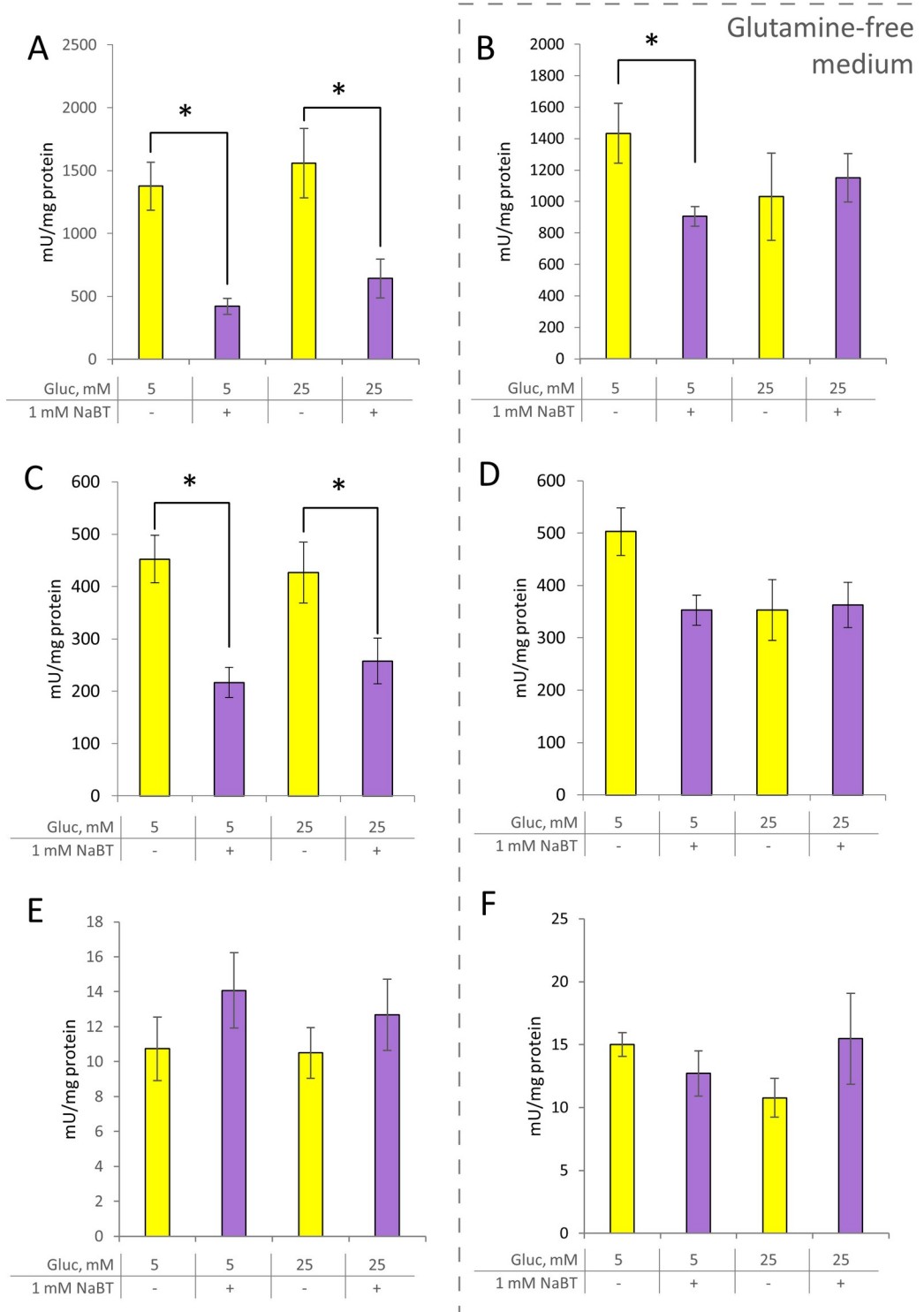

**Fig 6. Effect of sodium butyrate on the activity of main glycolytic enzymes.** (A, B): Lactate dehydrogenase activity (C,D): Pyruvate kinase activity. (E,F):Hexokinase activity. B,D,F–cells were grown in glutamine-free medium. All data are presented as mean ± SEM (n = 3–5; *p < 0.05, ANOVA followed by Turkey post hoc test). Gluc–glucose, NaBT–sodium butyrate.

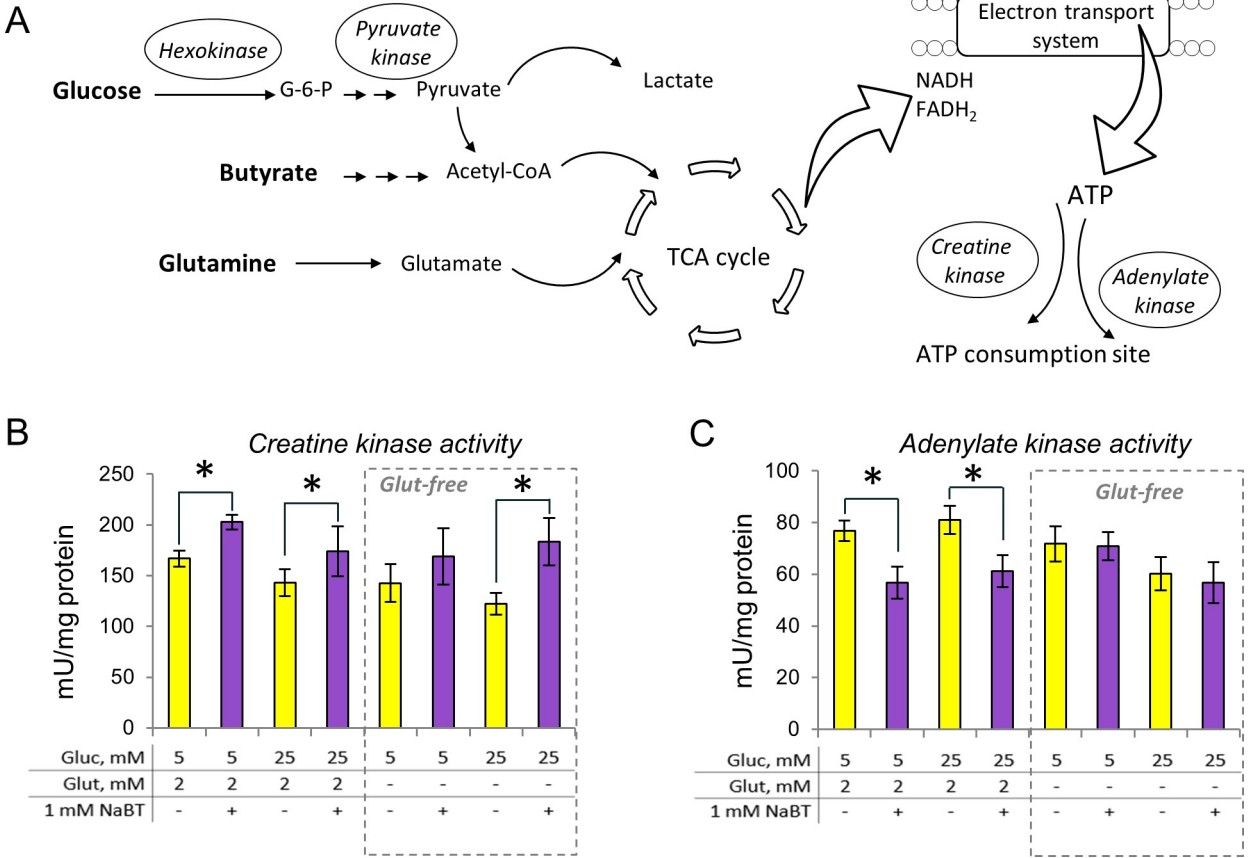

**Fig 7. Rearrangement of phosphotransfer system after treatment of Caco-2 cells with sodium butyrate.** (A): Schematic representation of main pathways analyzed. (B): Creatine kinase activity. (C): Adenylate kinase activity. All data are presented as mean ± SEM (n = 3–5; *p < 0.05, ANOVA followed by Tukey post hoc test). AK–adenylate kinase, CK–creatine kinase, G-6-P–glucose-6-phosphate, Gluc–glucose, Glut–glutamine, NaBT–sodium butyrate, TCA–tricarboxylic acid.

results suggested that in Caco-2 cells, butyrate affects later stages of glycolytic pathway, rather than earlier stages.

The activity of CK was increased and the activity of AK was decreased after differentiation with NaBT in the presence of glucose (5 mM) and glutamine. An increase in glucose concentration did not affect AK and CK activities (Fig 7B and 7C). The pattern of CK activity remained similar in glutamine-free media as the activities increased after cell incubation with NaBT regardless of glutamine or glucose availability in the growth medium. In contrast, the suppressive effect of NaBT on AK activity disappeared in glutamine-free media. Altogether, the results suggest that NaBT, in the presence of glutamine and glucose, can modulate cancer-induced changes in the phosphotransfer system.

## Discussion

Although conventional chemotherapy is the main treatment strategy for cancer, the lack of sensitivity and the development of resistance to currently available drugs limit significantly the efficiency of existent therapies [58]. Cancer stem cells, which demonstrate unique metabolic flexibility, are considered to be the reason for tumor relapse and metastasis and were shown to exhibit a remarkable resistance to chemo- and radiotherapy [59]. Reactivation of endogenous differentiation programs that would make cancer cells more amenable to treatment using

conventional therapeutic approaches appears to be one of the promising strategies for targeting CSCs in colon cancer [60]. Thus, the ability of butyrate and its derivatives to act as epigenetic modulators and promote differentiation of cancer cells has important implication for cancer prevention and therapy.

Currently, the identification of CSCs relies mostly on the expression of CSC surface markers. Numerous cell surface markers have been proposed for colon CSCs including CD133, CD44, LGR5, EpCAM, ALDH etc. The variability of CSC surface markers creates a necessity to identify more putative markers for colon CSCs. Factors regulating pluripotent stem cell renewal such as OCT4, SOX2 or NANOG, have potential for CSC identification [61,62], but one has to take into account the existence of different splice variants and expressed pseudogenes for OCT4 and NANOG. We aimed to characterize the expression of the main pluripotency-associated genes in colorectal cancer and distinguish it from the expression of relevant pseudogenes. In order to follow the differentiation status of butyrate-treated cells, we used alkaline phosphatase activity, which is a widely accepted differentiation marker for colon and colon cancer cells [2–6]. As quantitative RT-PCR is challenging and cannot be used sometimes to distinguish between different transcript variants or expressed pseudogenes, our study also includes semi-quantitative analysis of expressed transcripts. We aimed to reveal the major pattern and track possible changes in the expression of CSC markers at RNA level. Firstly, we used semi-quantitative densitometric measurements to quantify band intensities of respective transcripts and then chose targets, which showed differences between analyzed samples to be additionally validated by quantitative RT-PCR, if possible. We did not see any major differences in the expression of NANOG transcripts between butyrate treated and un-treated Caco-2 cells or between primary tumor and normal tissue samples. Apart from OCT4A/OCT4 pseudogenes and an increase in the expression of OCT4B4 transcript after butyrate treatment in Caco-2 cells, there were no apparent changes detected for OCT4B or OCT4B1. It was surprising to find out that NaBT induced OCT4A expression in Caco-2 cells. Along with this, transcription of OCT4A was mainly missing from tumor samples and instead elevated expression of OCT4 pseudogenes was detected, which coincides well with a previous study on colorectal cancer tissue [16]. Interestingly, decreased expression of SOX2 was noted in tumor samples compared with the respective control tissue samples, while there were no significant changes in the expression of SOX2 after NaBT treatment on Caco-2 cells. Previously, expression of SOX2 has been shown to be associated with CSC phenotype in colorectal cancer and lower levels of SOX2 expression have been suggested to associate prognosis for relapse-free survival [25] and forced expression of OCT4 and SOX2 has been shown to induce CSCs properties including sphere formation, chemoresistance and tumorigenicity [63]. Therefore, our results showing the opposite in tumor samples for SOX2 are challenging new studies to be conducted to find out more about SOX2 expression. Of note, we also found that some of the published SOX2 primers we tested were actually amplifying non-specific transcripts and we had to test several before choosing the ones that worked logically for human embryonic stem cells and cancer cells. Although rather speculative, our results showing that treatment with sodium butyrate induces OCT4A expression in differentiating colorectal Caco-2 cells could mark the phenotype of endodermal differentiation as it is known that induced expression of OCT4 can promote this type of differentiation under certain conditions [64]. Interestingly, it has been also noted that sodium butyrate supports endodermal differentiation [65–67]. Alternatively, the induced expression of OCT4A after NaBT treatment may reflect partial reprogramming in some of the cancer cells. The latter may not be preferable and should be further studied in the context of cancer stem cells, differentiation and cell survival. As the analyzed primary tumor samples expressed transcripts from pseudogenes and at the same time it was possible to induce OCT4A expression with sodium butyrate in differentiating Caco-2 cells, it gives OCT4A a new

possible role in colorectal cancer cells regulating the differentiation state and underscores the need of careful distinguishing between OCT4A and the relevant pseudogenes in future studies.

Enormous metabolic plasticity of cancer cells allows them to survive in severe environmental conditions that would be harmful to most normal cells. Cancer cells of the same tumor might use predominantly glycolysis or OXPHOS for energy production depending on nutrient availability within a given intertumoral compartment [68]. In contrast to normal cells, cancer cells often enhance glycolytic flux that supports cells with sufficient level of energy and intermediates for the biosynthetic pathways required for rapidly proliferating cells [69]. In the current paper, we show that about 70% of ATP produced by Caco-2 cells is derived from glycolysis while only 30% comes from the OXPHOS. Treatment of cells with NaBT allowed us to shift energy metabolism towards more oxidative metabolism, where butyrate is used as a preferred substrate for OXPHOS. Similar results were previously obtained using lung, breast and colon cancer cell lines [32–36]. Moreover, we found that butyrate not only increase OXPHOS but also non-mitochondrial residual oxygen consumption. Increased non-mitochondrial oxidation may be related with elevated intracellular levels of reactive oxygen species (ROS). it was demonstrated that butyrate can increase ROS levels more than 40% in hepatocellular carcinoma cells [70]. There are two possible mechanism how can butyrate effect on intracellular ROS level. Firstly, by decreasing flux through pentose phosphate shunt as a result decrease NADPH level [35,36,71]. Secondly, by alteration of Thioredoxin-1 expression [72]. In addition, our experiments revealed reduced glycolytic flux in Caco-2 cells after treatment with NaBT. Unaltered HK activity and decreased PYK and LDH activities suggested that NaBT affected the later stages of glycolytic pathway, rather than the earlier steps. These results are in contrast with studies by Amoêdo et al, who observed significantly increased HK activity and unchanged PYK and LDH activities after treatment of H460 lung cancer cells with NaBT [32]. Consequently, we assume that different cancer cell lines might respond in an individual manner to NaBT depending on the metabolic phenotype of utilized cell type; this has been also pointed out by other investigators [34].

In our previous studies, we observed the rearrangement of phosphotransfer network induced by malignant transformation of cancer cells [38–40]. Interestingly, increased AK and decreased CK activities were observed in cancer compared to control samples [38–40]. Here, we show that treatment of Caco-2 cells with NaBT may reverse cancer-induced changes in the phosphotransfer system as treated cells displayed increased CK activity and decreased AK activity. One possible way how butyrate may affect phosphotransfer network involves activation of AMP-dependent kinase (AMPK). Butyrate has been reported to modulate the energy metabolism of the cell by altering the phosphorylation status of AMPK directly or indirectly [73]. The AMPK can be activated by changes in the intracellular ATP/ADP ratio. Once activated, AMPK stimulates catabolic pathways to generate ATP, while turning off energy-consuming anabolic pathways [74]. In the cell, the level of ATP and ADP is monitored by the AK that catalyzes the conversion of ADP to ATP and AMP. Because of the AK reaction, even slight changes in ATP concentration lead to the drastic increase of cellular AMP concentration resulting in the AMPK activation [75]. Whether altered AK network in NaBT-treated cells is a consequence or a cause of AMPK activation needs to be determined.

In the current work, we showed that the response of phosphotransfer system to the butyrate treatment is modulated by the availability of glutamine in the growth medium. Glutamine is an important substrate and signaling molecule for intestinal epithelial proliferation and colonic crypt expansion [76]. In addition, glutamine may serve as an alternate fuel for cancer cells [68]. However, it is unknown whether the availability of different energy sources including glutamine and butyrate may alter phosphotransfer network of the cell. This study provides the first step toward clarification of this gap. Although the CK network responded to NaBT-

treatment similarly in the presence and in the absence of glutamine, notable changes were observed in the AK activity. The suppressive effect of NaBT on the AK activity disappeared during glutamine deprivation. The supplementation of glutamine-free medium with additional glucose resulted in decreased AK activity in both NaBT-treated and untreated Caco-2 cells. Future studies are required to define how AK network interferes with glucose and glutamine metabolism. In addition, the suppressive effect of NaBT on PYK and LDH activities was also abolished after treatment of cells with additional glucose in the absence of glutamine. A potential explanation for this phenomenon is the adaptation of cellular energy pathways to glutamine deprivation. The removal of glutamine from the growth medium alters cellular and mitochondrial NADH and $NAD^+$ ratios [77]. In order to cope with emerged imbalance, cancer cells may keep high glycolytic rate including high PYK and LDH activities. Collectively, our results link rearrangements in phosphotransfer network with metabolic alterations induced by NaBT treatment.

To our knowledge, this is the first study to describe the alterations in the phosphotransfer network induced by butyrate treatment of colon cancer cells. Our data indicate a close interplay between the phosphotransfer networks and metabolic plasticity of CRC, which is associated with the cell differentiation state. Further studies are needed to clarify the mechanisms behind the butyrate-mediated changes in AK network and its dependence on the glutamine and glucose availability.

## Supporting information

**S1 Fig. Schematic depiction of the oxygraphy protocols employed in the current study.** (PPTX)

**S2 Fig. The effect of NaBT on the survival of Caco-2 cells in trypan blue excluding test: Exposure time was 48 hours.** Bars are SEM (n = 3); ***$p<0.001$ (Student's t test). (PPTX)

**S3 Fig. Expression of pluripotency-associated transcription factors in Caco-2 cells after 48h treatment with 1 mM sodium butyrate.** (A): Detection of main OCT4 spliced variants, transcripts from NANOG1, NANOG2 and SOX2. GAPDH was used for loading control. OCT4A* primers can also amplify transcripts from OCT4 pseudogenes. OCT4B/B1 primers allow detection of OCT4 variant OCT4B4* (B): Restriction analysis of OCT4A* PCR product with ApaI showing 204 bp and 291bp fragments in the presence of OCT4A and 496bp product representative of OCT4 pseudogenes. (TIF)

**S4 Fig. Expression of pluripotency-associated transcription factors in primary colorectal tumor (T) and adjacent tissue (Ctrl) samples.** (A): Detection of main OCT4 spliced variants, transcripts from NANOG1, NANOG2 and SOX2. GAPDH was used for loading control. OCT4A* primers can also amplify transcripts from OCT4 pseudogenes. OCT4B/B1 primers allow detection of OCT4 variant OCT4B4* (B): Restriction analysis of OCT4A* PCR product with ApaI showing 204 bp and 291bp fragments in the presence of OCT4A and 496bp product representative of OCT4 pseudogenes. (TIF)

**S5 Fig. OCT4B/B1 primers allow detection of OCT4 variant OCT4B4. OCT4B1 product is 492 bp, OCT4B is 267 bp and OCT4B4 product is 239 bp.** OCT4B1 is highly expressed in human embryonic stem cells (hESC) together with both OCT4B and OCT4B4 transcripts. Caco-2 cells express OCT4B1 and OCT4B4, but not the OCT4B variant. In colorectal tumor and respective control samples the expression of OCT4B4 and OCT4B varies while OCT4B1 is

expressed on low level.
(PPTX)

**S6 Fig. Effect of sodium butyrate on the mitochondrial mass and membrane potential.** (A): Specific citrate synthase activity. (B,C): Mitochondrial membrane potential was assayed by TMRE and MitoTracker Red staining using fluorescence microplate reader. Data presented as mean ± SEM (n = 5). NaBT–sodium butyrate.
(PPTX)

**S1 Table. Changes in the AK isoforms in cancer cells and tissues described in the literature.**
(DOCX)

**S2 Table. Primers and product sizes in quantitative RT-PCR.**
(DOCX)

**S3 Table. Quantitative RT-PCR amplification program.**
(DOCX)

**S4 Table. Primers and product sizes in semi-quantitative RT-PCR.**
(DOCX)

**S5 Table. Semi-quantitative RT-PCR amplification programs.**
(DOCX)

**S1 File.**
(PDF)

**S1 Raw images.**
(PDF)

## Author Contributions

**Conceptualization:** Ljudmila Klepinina, Aleksandr Klepinin, Martin Pook.

**Data curation:** Ljudmila Klepinina, Aleksandr Klepinin.

**Formal analysis:** Ljudmila Klepinina, Aleksandr Klepinin, Laura Truu, Kaisa Kuus, Indrek Teino, Martin Pook.

**Funding acquisition:** Tuuli Kaambre.

**Investigation:** Ljudmila Klepinina, Indrek Teino, Martin Pook.

**Methodology:** Ljudmila Klepinina, Aleksandr Klepinin, Heiki Vija, Kaisa Kuus, Indrek Teino, Martin Pook.

**Resources:** Toivo Maimets, Tuuli Kaambre.

**Validation:** Ljudmila Klepinina, Aleksandr Klepinin.

**Writing – original draft:** Ljudmila Klepinina, Vladimir Chekulayev, Martin Pook.

**Writing – review & editing:** Ljudmila Klepinina, Aleksandr Klepinin, Laura Truu, Vladimir Chekulayev, Indrek Teino, Martin Pook, Toivo Maimets, Tuuli Kaambre.

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
