## [Decision Letter · Decision Letter 0]

2 Oct 2020

Pécs, Hungary

October 2, 2020

PONE-D-20-18387

Linking phosphotransfer network and metabolic plasticity with the differentiation status of colon cancer (Caco-2) cells

PLOS ONE

Dear Dr. Klepinina,

Thank you for submitting your manuscript to PLOS ONE. After careful consideration, we feel that it has merit but does not fully meet PLOS ONE’s publication criteria as it currently stands. Therefore, we invite you to submit a revised version of the manuscript that addresses the points raised bx the Reviewers, listed below.

We look forward to receiving your revised manuscript.

Kind regards,

Joseph Najbauer, Ph.D.

Academic Editor

PLOS ONE

Journal Requirements:

3. Please provide additional information about each of the cell lines used in this work, including any quality control testing procedures (authentication, characterisation, and mycoplasma testing). For more information, please see http://journals.plos.org/plosone/s/submission-guidelines#loc-cell-lines.

4. To comply with PLOS ONE submission guidelines, in your Methods section, please provide additional information regarding your statistical analyses, including the specific name and version of the software used and the threshold set for statistical significance in the analyses. For more information on PLOS ONE's expectations for statistical reporting, please see https://journals.plos.org/plosone/s/submission-guidelines.#loc-statistical-reporting.

5. In your Methods section, please provide additional information about the participant recruitment method and the demographic details of your participants. Please ensure you have provided sufficient details to replicate the analyses such as: a) the recruitment date range (month and year), b) a description of any inclusion/exclusion criteria that were applied to participant recruitment, c) a table of relevant demographic details, d) a statement as to whether your sample can be considered representative of a larger population and e) a description of how participants were recruited.

Reviewers' comments:

Reviewer's Responses to Questions

**Comments to the Author**

1. Is the manuscript technically sound, and do the data support the conclusions?

Reviewer #1: No

Reviewer #2: Yes

Reviewer #3: Partly

2. Has the statistical analysis been performed appropriately and rigorously? 

Reviewer #1: N/A

Reviewer #2: Yes

Reviewer #3: Yes

3. Have the authors made all data underlying the findings in their manuscript fully available?

Reviewer #1: Yes

Reviewer #2: Yes

Reviewer #3: Yes

4. Is the manuscript presented in an intelligible fashion and written in standard English?

Reviewer #1: Yes

Reviewer #2: Yes

Reviewer #3: Yes

5. Review Comments to the Author

Reviewer #1: Comments to Authors:

the study focuses on differentiation induction in the cancer cells, in figure 1, the authors show mild toxicity of around 10% which may be attributed to halt in cell cycle to undergo differentiation. However authors didnt show such markers, merely doing RT-PCR analysis of stem cell markers is not sufficient. Authors must show the cell surface marker analysis and also analyse the differentiated cells for their carcinogenic potential.

authors have covered the metabolic aspects of the NaBT treatment very well.

I suggest, if the authors want to show differentiation, merely moss of stem cell markers is not sufficient. the authors must characterize the differentiated cells, measure the cell cycle profiling, lookout for morphological changes in those cells etc.

Reviewer #2: The manuscript by Ludmila Klepanina and co-workers entitled „Linking phosphotransfer network and metabolic plasticity with the differentiation status of colon cancer (Caco-2) cells” describes the studies on the impact of sodium butyrate on colon adenocarcinoma cellular respiration in the context of de-differentiation. The incidence of colorectal cancer continues to increase, especially in developing countries, which is related to lifestyle changes leading to obesity, sedentary lifestyle, high red meat consumption, alcohol, and tobacco overuse. Despite the development of medicine, colorectal cancer is still the third most lethal cancer worldwide. It seems that an important role in the prevention of this neoplasm, as well as in the improvement of therapeutic strategies, is to better understand the mechanisms that guard the regulation of the correct physiology of the intestinal epithelium. It has been postulated for a long time that metabolites produced locally by the intestinal microflora are of significant importance in this process. One of them is butyric acid.

In this context, the studies undertaken by the authors of this paper seem to be justified and important from the point of view of the challenges of modern biomedicine. The authors studied the CaCo-2 cell line, which is a broadly used model of intestinal epithelial cell differentiation, to track the impact of sodium butyrate on pluripotency-regulatory genes expression, modulation of respiratory pathway and ATP metabolism.

I consider this research to be quite basic, but at the same time its results are interesting and can be a starting point for further, more mechanistic research. The goals of the experiments are quite clearly defined and clearly described. The English of the manuscript is correct and clear, without major linguistic and grammatical errors. The methodology is selected correctly and the experiments are conducted in the context of appropriately selected control samples. In general, the results are clearly expressed and discussed, however I wolud ask the authors to clarify or improve some issues:

1. The authors does not indicate in which cases a t-test and in which ANOVA was used for statistics.

2. I wolud recommend to present the results from Fig. 1 rather as a viability or cytotoxicity curves, not bar plots. This way of presentation of time-coursed data is far more legible.

3. I wolud suggest to present the S4 Fig and S5 Fig as a main figures instead of Fig 3 and Fig 4, respectively. The quantitative results, derived from PCR mesurements are of the main importance for the discussion and these are currently presented as a suplement. The pictures of bands in gels of all replicates are of course imporatnt, but more technical.

4. It is worth to clarify, why a specific PCR primers were not designed for OCT4B4 (line 306-307), similarly as it was done in the case of OCT4A?

Minor issues:

a) the scale bars on Fig 2 are not indicated with the unit;

b) in some figure captions there are the same p-values assigned for different markers (eg. Fig 1 „*p < 0.01; **p < 0.01);

c) The Introduction is quite wordy and I would consider a slight cut.

Reviewer #3: In this article, the authors analyzed the effect of sodium butyrate (NaBT) on the energy metabolism of Caco-2 cells coupled with its differentiation and changes in the phosphotransfer network. I found it really interesting, but some changes may improve the article.

1- In figure 1, the authors used the MTT assay to analyze cell viability. However, it should be considered that this is a metabolic assay, where the enzymatic reduction of 3- [4,5-dimethylthiazol-2-yl] -2,5-diphenyltetrazolium bromide (MTT) in MTT-formazan is catalyzed by succinate dehydrogenase, the complex II of the electron transport chain. Thus, the MTT assay depends on mitochondrial respiration, a step in which the NaBT appears to modulate, according to the authors. Given this, the authors could consider using other assays to analyze cell viability and/or cell proliferation, such as assays with ANEXIN and PI, and BrdU, respectively.

2- In the supplementary figure 5, the authors shown a quantitative RT-PCR analysis OCT4A, this graphic should not be supplementary. In fact, I believe that all semi-quantitative and quantitative data should be considered as main figures rather than supplementary.

3- Expression of SOX2 was more detected in normal samples and most tumor tissue samples had less SOX2 expressed than in adjacent normal tissue. Why does this happen? Protein expression could be analyzed.

4- In figure 5C, the authors shown an increased non- mitochondrial oxygen consumption after NaBT treatment. How is the production of reactive oxygen species in these cells?

5- The authors use figure 6 to describe 2 sets of results, with and without glutamine. I found it a bit confusing, because they are on the same graphics. Maybe if the graphics were divided and separated into 2 figures, it would be better.

6- The authors states that their results suggested that in Caco-2 cells, butyrate affects later stages of glycolytic pathway, rather than earlier stages. Could glucose be diverted to glycolytic shunts, such as the hexosamine pathway or pentose phosphate pathway? The entry of glucose into the pentose phosphate pathway, as glucose-6-phosphate, could explain the increased consumption of non-mitochondrial oxygen observed in figure 5C, since this pathway is important for the production of ROS.

7- Functional characteristics such as proliferation and migration are also associated with alterations in the phosphotransfer network induced by butyrate treatment of colon cancer cells? These functional characteristics and the role of changes in phosphotransfer network induced by NaBT could be analyzed.

8- Despite the title being "Linking phosphotransfer network and metabolic plasticity with the differentiation status of colon cancer (Caco-2) cells", the link is not very clear in the article. Perhaps the use of inhibitors and the subsequent analysis of differentiation markers, make this link clearer.

6. PLOS authors have the option to publish the peer review history of their article (what does this mean?). If published, this will include your full peer review and any attached files.

Reviewer #1: No

Reviewer #2: **Yes: **Paweł Link-Lenczowski

Reviewer #3: No

---

## [Author Response · Author response to Decision Letter 0]

9 Dec 2020

We thank the reviewers for their thoughtful review and constructive comments on the manuscript. The reviewers have raised important issues and we appreciate the opportunity to clarify our research objec-tives and results. Each comment has been carefully considered point by point, and we have revised our manuscript accordingly. Responses to the reviewers and changes in the revised manuscript are as follows.

Answers to Reviewer #1 comments

Rev#1: the study focuses on differentiation induction in the cancer cells, in figure 1, the authors show mild toxicity of around 10% which may be attributed to halt in cell cycle to undergo differentiation. However authors didnt show such markers, merely doing RT-PCR analysis of stem cell markers is not sufficient. Authors must show the cell surface marker analysis and also analyse the differentiated cells for their carcinogenic potential.

A: We agree that by measuring cell surface antigens associated with differentiation and performing additional analysis to estimate cell carcinogenic potential would add valuable information regarding the carcinogenic phenotype of the cells, but in our current work, we were more focused on the metabolic plasticity of those cells. As colorectal cancer is in general a heterogeneous population of cells with distinct differentiation levels, here we used a well-known general indicator, alkaline phosphatase activity, to estimate overall cell differentiation and the analysis of pluripotency-associated markers was additionally performed for indicating the presence of putative cancer stem cells. In order to better clarify how and why we used alkaline phosphatase activity for estimation of differentiation, we added relevant sentences and additional citations in the manuscript text. Furthermore, we modified Figure 1 by adding alkaline phosphatase activity measurement results from Supp. Fig. 3 to this main figure as this enables better understanding of the differentiation status of these cells. 

Rev#1: authors have covered the metabolic aspects of the NaBT treatment very well.

I suggest, if the authors want to show differentiation, merely moss of stem cell markers is not sufficient. the authors must characterize the differentiated cells, measure the cell cycle profiling, lookout for morphological changes in those cells etc.

A: In the current manuscript, we used a well-known general indicator of colon cancer cell differentiation, alkaline phosphatase activity, to estimate overall cell differentiation and the analysis of pluripotency-associated markers was additionally performed for indicating the presence of putative cancer stem cells. We also characterized the cell morphological changes associated with differentiation (Figure 2). We agree that further characterization of the differentiating cells would be a nice approach if to focus on cell tumorigenic potential. We will take this into account when we are planning next experiments for subsequent publications, but in the current manuscript we wanted to focus more on metabolic aspects.

Answers to Reviewer #2 comments

The manuscript by Ludmila Klepinina and co-workers entitled „Linking phosphotransfer network and metabolic plasticity with the differentiation status of colon cancer (Caco-2) cells” describes the studies on the impact of sodium butyrate on colon adenocarcinoma cellular respiration in the context of de-differentiation. The incidence of colorectal cancer continues to increase, especially in developing countries, which is related to lifestyle changes leading to obesity, sedentary lifestyle, high red meat consumption, alcohol, and tobacco overuse. Despite the development of medicine, colorectal cancer is still the third most lethal cancer worldwide. It seems that an important role in the prevention of this neoplasm, as well as in the improvement of therapeutic strategies, is to better understand the mechanisms that guard the regulation of the correct physiology of the intestinal epithelium. It has been postulated for a long time that metabolites produced locally by the intestinal microflora are of significant importance in this process. One of them is butyric acidIn this context, the studies undertaken by the authors of this paper seem to be justified and important from the point of view of the challenges of modern biomedicine. The authors studied the CaCo-2 cell line, which is a broadly used model of intestinal epithelial cell differentiation, to track the impact of sodium butyrate on pluripotency-regulatory genes expression, modulation of respiratory pathway and ATP metabolism.

I consider this research to be quite basic, but at the same time its results are interesting and can be a starting point for further, more mechanistic research. The goals of the experiments are quite clearly defined and clearly described. The English of the manuscript is correct and clear, without major linguistic and grammatical errors. The methodology is selected correctly and the experiments are conducted in the context of appropriately selected control samples. In general, the results are clearly expressed and discussed, however I wolud ask the authors to clarify or improve some issues:

1. The authors does not indicate in which cases a t-test and in which ANOVA was used for statistics.

A: In revised manuscript, we specified methods of statistical analysis in figure legends.

2. I wolud recommend to present the results from Fig. 1 rather as a viability or cytotoxicity curves, not bar plots. This way of presentation of time-coursed data is far more legible.

A: Figure 1 was modified as suggested.

3. I wolud suggest to present the S4 Fig and S5 Fig as a main figures instead of Fig 3 and Fig 4, respectively. The quantitative results, derived from PCR mesurements are of the main importance for the discussion and these are currently presented as a suplement. The pictures of bands in gels of all replicates are of course imporatnt, but more technical.

A: We rearranged the figures according to the suggestion in order to present the results based on the importance.

4. It is worth to clarify, why a specific PCR primers were not designed for OCT4B4 (line 306-307), similarly as it was done in the case of OCT4A?

A: The OCT4B4 transcript is not well characterized so far in different cell lines and the sequence is still not in NCBI's reference sequence (RefSeq) database. Therefore, we decided not to design the primers ourselves. We tried to use the primers published in the article of OCT4B4 discovery (Poursani et al. 2017), but in our hands these did not work. We also tried to contact the authors for additional information, but got no answer. However, we found that this transcript, which is characterized as very similar one to OCT4B can be detected with the OCT4B/B1 primers (Atlasi et al., 2008) we used for our semi-quantitative analysis and these results are supported by the findings in the article of OCT4B4 discovery (Poursani et al. 2017).

Minor issues:

a) the scale bars on Fig 2 are not indicated with the unit;

b) in some figure captions there are the same p-values assigned for different markers (eg. Fig 1 „*p < 0.01; **p < 0.01);

c) The Introduction is quite wordy and I would consider a slight cut.

A: The scale bars are indicated in Fig 2 legend. P values were corrected. The Introduction was made shorter as was suggested

Answers to Reviewer #3 comments

In this article, the authors analyzed the effect of sodium butyrate (NaBT) on the energy metabolism of Caco-2 cells coupled with its differentiation and changes in the phosphotransfer network. I found it really interesting, but some changes may improve the article.

1-In figure 1, the authors used the MTT assay to analyze cell viability. However, it should be considered that this is a metabolic assay, where the enzymatic reduction of 3- [4,5-dimethylthiazol-2-yl] -2,5-diphenyltetrazolium bromide (MTT) in MTT-formazan is catalyzed by succinate dehydrogenase, the complex II of the electron transport chain. Thus, the MTT assay depends on mitochondrial respiration, a step in which the NaBT appears to modulate, according to the authors. Given this, the authors could consider using other assays to analyze cell viability and/or cell proliferation, such as assays with ANEXIN and PI, and BrdU, respectively.

A: In the current study, we used viability assay (MTT) together with cytotoxic assay (LDH release) (Fig 1A, B) and trypan blue test (S2 Fig) to validate the best experimental conditions to study effects of NaBT on cellular bioenergetics. 1 mM concentration of butyrate was found appropriate to induce differentiation without any toxic effects. The concentration of butyrate was validated based not only on MTT but also on LDH release assay and trypan blue test. In addition, we have chosen MTT test to be sure that our cell line responds to butyrate treatment in the same manner as reported in previous works (Hyan Ri Kang et al., 2016, Xiaofei Sun and Mei-Jun Zhu 2018, LUYING PENG et al 2006). Indeed, it is a good suggestion to use ANEXIN (apoptosis detection kit), BrdU (cell proliferation assay) and Pi (cell cycle), and it has been also done by other groups. According to the literature, butyrate at higher concentrations (≥2 mM) deprives cell proliferation (Hyang Ri Kand et al 2016, Carmel Aviv-Green et al 2002), induces cell cycle arrest (Hyang Ri Kang et al 2016) and apoptosis via Caspase-3 pathway (Jintao Zhang et al 2015, ARKADIUSZ ORCHEL et al 2005, Carmel Aviv-Green et al 2002).

2- In the supplementary figure 5, the authors shown a quantitative RT-PCR analysis OCT4A, this graphic should not be supplementary. In fact, I believe that all semi-quantitative and quantitative data should be considered as main figures rather than supplementary.

We rearranged the figures according to the suggestion and the quantitative results are now presented as main figures (Fig 3 and 4).

3- Expression of SOX2 was more detected in normal samples and most tumor tissue samples had less SOX2 expressed than in adjacent normal tissue. Why does this happen? Protein expression could be analyzed.

This is a good question, but we don’t know why we detected less SOX2 in tumor samples while in case of Caco-2 cells we did not see any change after NaBT treatment. We agree that this needs further experiments to see how SOX2 could be regulated on protein level. Unfortunately, the amount of tissue samples was limited and we had to choose the assays to perform. Currently, we cannot obtain additional samples to look at the expression on protein level.

4- In figure 5C, the authors shown an increased non- mitochondrial oxygen consumption after NaBT treatment. How is the production of reactive oxygen species in these cells?

Yes, increased non-mitochondrial oxygen consumption after NaBT treatment should be related with ROS production. For example, it was demonstrated that butyrate is able to increase ROS levels more than 40 % in hepatocellular carcinoma cells (Kishor Pant et al., 2017). There are two pathways how butyrate is able to increase ROS production:

1) By decreasing flux through pentose phosphate shunt as a result decrease NADPH level

2) By alteration of Thioredoxin-1 expression.

See also question number 6 answer

5- The authors use figure 6 to describe 2 sets of results, with and without glutamine. I found it a bit confusing, because they are on the same graphics. Maybe if the graphics were divided and separated into 2 figures, it would be better.

To make results clearer, we separated graphics in figure 6 based on glutamine presence. 

6- The authors states that their results suggested that in Caco-2 cells, butyrate affects later stages of glycolytic pathway, rather than earlier stages. Could glucose be diverted to glycolytic shunts, such as the hexosamine pathway or pentose phosphate pathway? The entry of glucose into the pentose phosphate pathway, as glucose-6-phosphate, could explain the increased consumption of non-mitochondrial oxygen observed in figure 5C, since this pathway is important for the production of ROS.

Oppositely, there are evidence that colorectal cancer cells treatment with 1mM butyrate decreases pentose phosphate pathway (Qingran Li et al 2018). Moreover, butyrate can decrease NADPH (Jean-Marc Blouin et al 2010) and, as a result, increase ROS production (Krushna C. Patra and Nissim Hay 2014). There is another pathway which is also a source of ROS production outside the mitochondria. Recently, it was found that butyrate could downregulate Thioredoxin-1 expression, which leads to increased ROS production (Wang et al, 2020). Further studies are needed to figure out if those pathways are related to increased non-mitochondrial oxygen consumption induced by butyrate.

7- Functional characteristics such as proliferation and migration are also associated with alterations in the phosphotransfer network induced by butyrate treatment of colon cancer cells? These functional characteristics and the role of changes in phosphotransfer network induced by NaBT could be analyzed.

Indeed, butyrate inhibits the invasive potency of colorectal cancer cells, which has been demonstrated in vitro in several cell models (Zeng, H et al., 2005, Li, Q et al, 2017, Wang, W et al., 2020). The mechanisms of this phenomenon may be partially mediated by changes in the overall activity and expression profile of AK and CK isozymes. There is evidence that alterations in the phosphotransfer network is associated with increased proliferation and migration activity of cancer cells. In colorectal cancer cells overexpression of adenylate kinase isoform 6 (AK6) enhances invasion and metastasis activity of tumor cells (Yapeng Ji et al., 2017). Although in our previous study we demonstrated that creatine kinase network is downregulated in colorectal tumor patients (Kaldma et al., 2014), then recently work on mice model demonstrated that colon cancer metastatic cells are dependent on CKB-mediated intracellular phosphocreatine (Jia Min Loo et al., 2015). Nevertheless, in current study we more focused on how butyrate affects cancer cells bioenergetics via alteration of phosphotransfer network. It was demonstrated that CK network has important role in colon cells where CK not only participate in bioenergetics of epithelial cells but also has important function in intestine mucosa barrier formation (David Kitzenberg et al., 2016). It is well known that butyrate is the main energy source for normal colonocytes, butyrate also promotes proliferation of colonic epithelial cells, and improves intestinal barrier function (Gonçalves P et al. 2013). Since butyrate can alter the cellular metabolism of cancer cells, therefore we hypothesized that treatment with sodium butyrate (NaBT) may reverse cancer-induced changes in phosphotransfer network of colon adenocarcinoma by restore colon epithelial cells homeostasis. This knowledge which we have got in current study can be used in future studies to understand how alterations in the phosphotransfer network induced by butyrate treatment of colon cancer cells are associated with suppression of cancer cells proliferation and migration activity.

8- Despite the title being "Linking phosphotransfer network and metabolic plasticity with the differentiation status of colon cancer (Caco-2) cells", the link is not very clear in the article. Perhaps the use of inhibitors and the subsequent analysis of differentiation markers, make this link clearer.

In current manuscript we characterized the differentiation status of Caco-2 cells by measuring the activity of alkaline phosphatase, which was increased after sodium butyrate treatment. We agree that further charaterization of additional differentiation markers along with inhibitor analysis would be interesting, but were currently not in our focus. We chose the widely used Caco-2 cell line to be firstly characterized for the pluripotency-associated stem cell markers along with the general estimation of cell differentiation by alkaline phosphatase activity. Subsequent experiments involving also different colon cancer cell lines could reveal better picture how various possible differentiation markers and the expression of pluripotency-associated markers are correlating during differentiation, but are out of scope of current study. We also agree that the “link” may not be so clear yet and we decided to change the title accordingly: „Colon cancer cell differentiation by sodium butyrate modulates metabolic plasticity of Caco-2 cells via alteration of phosphotransfer network”

---

## [Decision Letter · Decision Letter 1]

29 Dec 2020

Pécs, Hungary

December 28, 2020

Colon cancer cell differentiation by sodium butyrate modulates metabolic plasticity of Caco-2 cells via alteration of phosphotransfer network

PONE-D-20-18387R1

Dear Dr. Klepinina,

We’re pleased to inform you that your manuscript (R1 version) has been judged scientifically suitable for publication and will be formally accepted for publication once it meets all outstanding technical requirements.

PLEASE NOTE: Fig. 2, please change MitoTraker Red to MitoTracker Red.

Kind regards,

Joseph Najbauer, Ph.D.

Academic Editor

PLOS ONE

Reviewers' comments:

Reviewer's Responses to Questions

**Comments to the Author**

1. If the authors have adequately addressed your comments raised in a previous round of review and you feel that this manuscript is now acceptable for publication, you may indicate that here to bypass the “Comments to the Author” section, enter your conflict of interest statement in the “Confidential to Editor” section, and submit your "Accept" recommendation.

Reviewer #2: All comments have been addressed

Reviewer #3: All comments have been addressed

2. Is the manuscript technically sound, and do the data support the conclusions?

Reviewer #2: Yes

Reviewer #3: Yes

3. Has the statistical analysis been performed appropriately and rigorously? 

Reviewer #2: Yes

Reviewer #3: Yes

4. Have the authors made all data underlying the findings in their manuscript fully available?

Reviewer #2: Yes

Reviewer #3: Yes

5. Is the manuscript presented in an intelligible fashion and written in standard English?

Reviewer #2: Yes

Reviewer #3: Yes

6. Review Comments to the Author

Reviewer #2: The authors fully answered my doubts and introduced the suggested changes. I find the results interesting and worth further, more mechanistic studies.

Reviewer #3: (No Response)

7. PLOS authors have the option to publish the peer review history of their article (what does this mean?). If published, this will include your full peer review and any attached files.

Reviewer #2: No

Reviewer #3: No

---

## [Editor Report · Acceptance letter]

11 Jan 2021

PONE-D-20-18387R1 

Colon cancer cell differentiation by sodium butyrate modulates metabolic plasticity of Caco-2 cells via alteration of phosphotransfer network 

Dear Dr. Klepinina:

I'm pleased to inform you that your manuscript has been deemed suitable for publication in PLOS ONE. Congratulations! Your manuscript is now with our production department. 

Kind regards, 

on behalf of

Dr. Joseph Najbauer 

Academic Editor

PLOS ONE